# Development and Characterization of Folic Acid-Conjugated Amodiaquine-Loaded Nanoparticles–Efficacy in Cancer Treatment

**DOI:** 10.3390/pharmaceutics15031001

**Published:** 2023-03-20

**Authors:** Vineela Parvathaneni, Snehal K. Shukla, Vivek Gupta

**Affiliations:** Department of Pharmaceutical Sciences, College of Pharmacy and Health Sciences, St. John’s University, Queens, NY 11439, USA

**Keywords:** amodiaquine, folic acid, cancer, nanoparticles, targeted delivery, 3D spheroid studies, repurposing, nano-repurposing

## Abstract

The objective of this study was to construct amodiaquine-loaded, folic acid-conjugated polymeric nanoparticles (FA-AQ NPs) to treat cancer that could be scaled to commercial production. In this study, folic acid (FA) was conjugated with a PLGA polymer followed by the formulation of drug-loaded NPs. The results of the conjugation efficiency confirmed the conjugation of FA with PLGA. The developed folic acid-conjugated nanoparticles demonstrated uniform particle size distributions and had visible spherical shapes under transmission electron microscopy. The cellular uptake results suggested that FA modification could enhance the cellular internalization of nanoparticulate systems in non-small cell lung cancer, cervical, and breast cancer cell types. Furthermore, cytotoxicity studies showed the superior efficacy of FA-AQ NPs in different cancer cells such as MDAMB-231 and HeLA. FA-AQ NPs had better anti-tumor abilities demonstrated via 3D spheroid cell culture studies. Therefore, FA-AQ NPs could be a promising drug delivery system for cancer therapy.

## 1. Introduction

In recent years, significant advances in the field of cancer therapeutics have immensely improved patient survival and quality of life. While an extensive amount of work is being conducted in the anti-cancer drug development domain, as demonstrated by 207 anticancer FDA drug approvals between 2016 and 2021, the majority of drugs (71%) were approved as add-on therapies or later-line therapies. This trend clearly demonstrates our lack of ability to develop new molecules capable of targeting novel pathogenic pathways [1,2]. Although immunotherapies have become the major player in the oncology domain, targeted small molecule chemotherapy remains the mainstay for cancer therapeutics. However, the current chemotherapeutic regimen suffers from major limitations including acquired resistance, adverse events/toxicity effects, and poor targetability [2,3]. In addition, the current chemotherapy often requires a combination treatment strategy using support therapies to tackle side effects of chemotherapeutic agents. Overall, this strategy is expensive, making it difficult for several patients to afford it [4]. This has raised the attention of many researchers to explore new drugs or new therapies for an effective cancer therapy [5]. As the discovery or development of new drugs may take exorbitantly long times, drug repurposing has become an attractive strategy to discover new uses of old FDA-approved drugs [6]. Hence, the utilization of old drugs against new indications in cancer therapeutics could be an invaluable approach toward novel discoveries and effective therapeutics.

In earlier work from our group, we demonstrated the feasibility of drug repurposing for various cancers’ treatments, together with non-small cell lung cancer (NSCLC), breast cancer, and mesothelioma [7,8,9,10]. We have shown that previous FDA-approved drugs can be successfully repurposed for the treatment of various aggressive cancer phenotypes with great success [7,11,12]. However, most repurposed therapies work by poly-pharmacology, and may have adverse effects in healthy tissues if given by systemic routes [13]. Further investigation into targeting the repurposed therapies to localize the therapy in the tumor vicinity is imperative for enabling their further development.

Most of the cancer types share many features including uncontrolled cell growth and a probability for metastasis [14]. In addition, most cancer cells exhibit an overexpression of certain receptors such as transferrin (Tf), folate (FRα), etc., which play a crucial role in cell functions, cellular proliferation (Tf), and regulating the cellular uptake of folate molecules (FRα) [15,16]. Targeting these receptors on cancer cells through specific targeting ligands is advocated for achieving a superior anti-cancer efficacy [16,17]. A cell surface receptor (FRα) for folic acid (FA) has been known to be overexpressed in the tumor cells of many human cancers such as non-small cell lung cancer, cervical, ovarian, and breast cancers [18]. Targeting FRα receptors through FA offers many potential advantages pertaining to its small size and promising pharmacokinetic properties of the FA conjugates. Moreover, this targeting ligand could reduce the probability of immunogenic reactions, allowing for repeated administration; this may offer simple and well-defined conjugation chemistry, enabling easier conjugation reactions [19]. FA receptors are capable of binding to folic acid, which is relatively innocuous, able to penetrate through solid tumors, and is feasible to chemically conjugate with therapeutic moieties and polymers [20]. In normal tissue, folic acid receptors are expressed at low levels, primarily on the apical surface of polarized cells, avoiding contact with blood circulation [21]. As the expression of folate receptors is highly restricted in normal tissues, the presence of a high receptor affinity of FA results in its high tumor tissue specificity [22]. FA binds to the FA receptors with high affinity and can target conjugated payloads, specifically cells with an overexpression of FA receptors [23]. Further internalization of the FA receptor–ligand complex takes place via endocytosis [19]. This process involves four steps: (1) FA binding to folate receptor, (2) internalization of the FA receptor-ligand complex, (3) dissociation of FA and movement into the cytoplasm, and (4) covalent addition of glutamic acid residues to FA [24].

Although nanocarriers have been known to enable the delivery of therapeutics effectively, the conjugation of targeting moieties would be of great importance in achieving a safer drug delivery. Several nanocarrier systems such as liposomes, polymeric nanoparticulate systems, mesoporous silica nanoparticles, gold nanoparticles, solid lipid nanoparticles, carbon nanotubes, protein nanoparticles, core shell nanoparticles and magnetic nanoparticles have been developed to achieve selective targeting against tumors [25,26]. Utilizing targeted nanocarriers in drug delivery establishes a platform for efficient, tumor-specific delivery by interacting with receptors on the surface of cancer cells [27].

The advantage of targeting FA receptors was initially exploited by Rosenholm et al. when they reported on the synthesis of folate-conjugated porous hybrid silica nanoparticles for targeting cancer cells [28]. Mesoporous silica nanoparticles functionalized by the surface hyperbranching polymerization of poly(ethylene imine), PEI, were further modified by introducing both fluorescent and targeting moieties, with the aim of specifically targeting cancer cells [28]. Liu et al. reported on the synthesis of FA-conjugated nanoparticles of mixed lipid monolayer shells and biodegradable polymer cores for the targeted delivery of docetaxel [29]. A system of nanoparticles of mixed lipid monolayer shells and biodegradable polymer cores was developed for the targeted delivery of anticancer drugs with Docetaxel as the model drug, which provides targeting versatility with a quantitative control of the targeting effect by adjusting the lipid component ratio of the mixed lipid monolayer, and combines the advantages and avoids the disadvantages of polymeric nanoparticles and liposomes in drug delivery [29]. Recently, Cheung et al. highlighted the importance of FA receptor targeting by summarizing the role of FA receptors with a special focus on treatment approaches [30]. Several other studies have investigated the targeting potential of drug-FA conjugates against different cancers, as summarized by Lutz et al. [23]. PLGA-PEG-FA-conjugated delivery systems have been utilized in several reports due to their superior targeting characteristics [31,32]. Similarly, the inclusion of a polyethylene glycol (PEG) spacer is a popular strategy for increasing hydrophilicity and avoiding aggregation. In addition, PEGylation provides passive tumor-targeting via enhanced permeability and retention effects. These systems are recognized selectively by the folic acid receptors in tumor cells to acquire receptor-mediated active tumor targeting [33].

Previous investigations from our group have established am FDA-approved anti-malarial drug, amodiaquine (AQ), as an effective anti-cancer therapeutic against a variety of cancers. AQ has been reported for its ability to inhibit autophagy, to block cell cycle progression, and to induce apoptosis in different cancer types among several studies from our research group [7,9,10]. Recently, the superior anti-cancer efficacy of AQ-loaded nanoparticles against NSCLC was also reported in a research study published from our group [7]. For the present study, we chose to investigate AQ for superior efficacy through a combination of nanoparticle encapsulation and FA receptor targeting. PLGA-based FA-conjugated NPs of AQ were formulated using a scalable, reproducible high-pressure homogenization (HPH) approach, and were evaluated for their anti-cancer efficacy. In the current project, we aim to achieve a further tumor targeting efficacy through FA-decorated nanoparticles.

In the present study, the anticancer activity of FA-conjugated amodiaquine-loaded nanoparticles (FA-AQ NP) against FA-receptors expressing cancer cells was evaluated. The process of functionalization/conjugation refers to the chemical addition of targeting ligand, FA, to enhance the target nanoparticles to the lung tumor tissue. AQ-loaded NPs with FA ligands were prepared and characterized by various analytical techniques for their physicochemical properties, and cytotoxic and targeting capabilities. Several 2D and 3D cell culture studies were carried out to assess the specificity of the folate receptors and to determine whether AQ-loaded and targeted nanoparticles exhibited better anti-cancer efficacy than the plain drug. The targeted nanoparticles were evaluated to verify the hypothesis that FA-decorated NPs can demonstrate a superior efficacy in treating multiple cancers.

## 2. Materials and Methods

### 2.1. Materials

RESOMER^®^ RG 502 (bioresorbable poly(d,l-lactide-co-glycolide) (50:50)) was obtained from Evonik Industries (Darmstadt, Germany). Polyvinyl alcohol (PVA) and AQ were purchased from Sigma-Aldrich (St. Louis, MO, USA). FA was obtained from Fisher Scientific (Hampton, NH, USA). Heat-inactivated fetal bovine serum (FBS) was obtained from Atlanta Biologicals (R&D Systems, Minneapolis, MN, USA). The A549, HeLa, MDAMB-231, and HEK-293 cell lines were obtained from ATCC (Manassas, VA, USA). The A549 cell line was maintained in an RPMI-1640 medium supplemented with 10% FBS, sodium pyruvate, and penicillin/streptomycin (Corning, NY, USA) at 5% CO_2_/37 °C. The HEK-293, HeLa and MDAMB-231 cell lines were maintained in a DMEM medium with FBS (10%) and penicillin/streptomycin 5% CO_2_/37 °C. Other cell cultures and analytical reagents were obtained from Fisher Scientific (Hampton, NH, USA). The molecular biology kits and antibodies were purchased from other sources, mentioned at respective methods.

### 2.2. UPLC Method Development

The UPLC method was established for quantifying AQ using Waters Acquity (Waters Corp., Milford, MA, USA), following a recently published method summarized in the Appendix A [7].

### 2.3. Calibration Curve for Folic Acid

Standard solutions (15.6, 31.3, 125, and 250 µg/mL) of FA were prepared by diluting with DMSO. The absorbance was measured at 256 nm using a plate reader (Tecan Spark 10M; Tecan Group Ltd., Männedorf, Switzerland). The calibration curve was constructed between the FA concentrations and absorbance units.

### 2.4. PLGA–PEG–FA Conjugate Synthesis

FA conjugation was performed following well-established EDC-NHS chemistry, reported previously with few modifications [34]. A brief synthetic scheme for the PLGA–PEG–FA conjugate preparation is presented in Figure 1B. Briefly, the PLGA solution (1 g/8 mL DCM) was activated by 80 mg of EDC-I and 57.5 mg of NHS at room temperature for 24 h under gentle stirring. The obtained solution was filtered and precipitated by adding in ice-cold diethyl ether and the resulting product of activated PLGA was entirely dried under a vacuum. Then, the organic solution of activated PLGA (0.5 g/4 mL DCM) was added slowly to the PEG–bis-amine solution (50 mg/1 mL DCM) in a dropwise manner. This reaction was carried out for 6 h under room temperature, after which time the resultant solution was precipitated after adding ice-cold diethyl ether. The precipitated product, amine-terminated di-block copolymer, PLGA–PEG–NH_2_ was filtered and dried. The FA-conjugated di-block copolymer was synthesized by coupling the PLGA–PEG–NH_2_ di-block copolymer with an activated FA. Briefly, 250 mg of di-block copolymer dissolved in 2.5 mL of dimethyl sulfoxide (DMSO) was mixed with 6.5 mg of FA and 6.5 mg of DCC. The reaction was carried out at room temperature for 7 h and then mixed with 50 mL of the cold methanol and filtered through the paper filter. The precipitate on the filter was dried under vacuum, and then dissolved in 25 mL of DCM. This way, the free FA was precipitated in DCM, but the conjugated FA was dissolved. After centrifugation at 21,000× *g*, the supernatant was dried under a vacuum. The FA-conjugated PLGA collected after centrifugation was analyzed for conjugation efficiency and was utilized in the formulation of nanoparticles.

### 2.5. Determination of Conjugation Efficiency

The percent of conjugation was calculated by determining the amount of FA conjugated in PLGA–PEG–FA. A known amount (0.59–0.81 mg) of dried PLGA–PEG–FA was dissolved in DMSO and an UV absorbance value at 256 nm was evaluated to determine the concentration of conjugated FA.

### 2.6. Formulation of Non-Targeted (AQ NP Rota) and FA-Targeted AQ NPs (FA-AQ NPs)

Non-targeted and FA-conjugated AQ nanoparticles were formulated through a scalable high-pressure homogenization (HPH) method, recently published and validated by our group, with slight modifications [9] (Figure 1C). Briefly, AQ-loaded nanoparticles (AQ NPs Rota) were prepared using 20 mg/mL PLGA 502H solution in DCM, AQ aqueous solution (5 mg/0.5 mL), and a stabilizing agent (20 mL of 1% *w/v* PVA solution in PBS). The FA-conjugated NPs (FA-AQ NPs) were prepared with a combination of both PLGA and PLGA-PEG-FA (FA-conjugated PLGA) at a 5:1 ratio, while the other formulation components were the same as for the non-targeted AQ-loaded NPs. A stable pre-emulsion was formed through probe homogenization at 25,000 rpm for 10 min, which was processed through Nano DeBee^®^ HPH (BEE International, South Easton, MA, USA) at 30,000 psi and 7 reverse flow cycles, as reported in our recent publication [7]. The formulation protocols are described in detail in the Appendix A.

### 2.7. Physicochemical Characterization: Particle Size, PDI, and Zeta Potential

The physicochemical properties such as the nanoparticle size, zeta potential, and polydispersity index of AQ NPs Rota and FA-AQ NPs were measured using Malvern^®^ zeta-sizer, following previously established protocols [9].

### 2.8. Drug Content

Following the removal of unencapsulated AQ, the loaded drug amount in nanoparticle formulation was evaluated utilizing a direct particle lysis methodology. In brief, 1980 µL of ACN:water:DCM–98:1.5:0.5 was added to 20 μL of nanoparticles, followed by centrifugation for 45 min at 4 °C at 21,000× *g* to lyse the nanoparticles and to acquire the loaded drug into the evaluating solution. The obtained supernatant was analyzed for the drug content using UPLC and the % EE/% DL was calculated.
(1)$$\%\;\mathrm{Entrapment}\;\mathrm{efficiency}\;\left(\%\mathrm{EE}\right)=\left[\frac{\mathrm{Drug}\;\mathrm{entrapped}\;\mathrm{in}\;\mathrm{nanoparticles}}{\mathrm{Total}\;\mathrm{drug}\;\mathrm{added}\;\mathrm{initially}}\right]\times 100$$
(2)$$\%\;\mathrm{Drug}\;\mathrm{loading}\;\left(\%\mathrm{DL}\right)=\left[\frac{\mathrm{Drug}\;\mathrm{entrapped}\;\mathrm{in}\;\mathrm{nanoparticles}}{\mathrm{Total}\;\mathrm{polymer}+\mathrm{drug}\;\mathrm{added}}\right]\times 100$$

### 2.9. Morphological Analysis

The imaging of unconjugated and conjugated AQ nanoparticles was performed using transmission electron microscopy (TEM) for evaluating nanoparticles’ morphology using 5 µL of the diluted nanoparticles’ sample. TEM imaging was performed using FEI Tecnai Spirit TWIN TEM (FEI, Hillsboro, OR, USA), operated at 120 kV voltage. The detailed procedure is described in the Appendix A.

### 2.10. Solid-State Characterization Studies

Powder X-ray Diffraction (PXRD): X-ray diffraction spectroscopy was employed using XRD-6000 (Shimadzu, Kyoto, Japan). The detailed methods are provided in the Appendix A.

Differential Scanning Calorimetry (DSC): The calorimetric data (thermograms) for AQ, AQ NP Rota, and FA-AQ NP were obtained by a closed pan approach employing a DSC 6000 (PerkinElmer; Shelton, CT, USA) equipped with an intra-cooler accessory. The detailed methods are provided in the Appendix A.

### 2.11. Stability of FA-AQ NPs

The stability of FA-AQ NPs was determined via storing the formulations (n = 3) at temperatures of 4 °C and 25 °C for a 4-week period, as reported earlier [35]. An aliquot of each sample was withdrawn at the end of weeks 1, 2, 3, and 4; diluted with water (100-fold); and analyzed for physicochemical characteristics such as particle size, PDI, and zeta potential using Malvern Zetasizer, as described earlier. The drug content was determined by lysing the samples using UPLC, as described in Section 2.8.

### 2.12. Determination of Targeting Capability of FA-AQ NPs

#### 2.12.1. Cellular Uptake (Microscopic Assessment)

Cellular uptake studies were carried out utilizing a protocol reported previously [36]. NPs were prepared by replacing AQ with coumarin-6 (C-6, 1 mg), a fluorescent dye commonly used in intracellular uptake studies which offers simple and sensitive detection for easier tracking [12,37,38]. The detailed description of the study protocol and imaging procedure are provided in the Appendix A.

#### 2.12.2. Cellular Uptake (Quantitative Assessment by Determining Fluorescence Intensity)

The quantitative assessment of intracellular uptake was performed in 3 different cancer cell lines, A549, HeLa, and MDAMB-231, as reported in our previous study [9]. The details are provided in the Appendix A.

### 2.13. Cytotoxicity Studies

AQ NP Rota and FA-AQ NPs, along with AQ were examined for their cytotoxicity in multiple cancer cell lines: A549 (non-small cell lung cancer, NSCLC); HeLa (cervical cancer), and MDAMB-231 (breast cancer), as reported earlier with slight modifications [8]. The HeLa and MDAMB-231 cell lines were chosen because of their ability to overexpress FA receptors [18]. Further cytotoxicity studies were executed on human embryonic kidney (HEK-293) cell lines to ascertain the safety of blank (drug-free) nanoparticles at 0.39, 0.78, 1.56, 3.13, 6.25, 12.5, 25 and 50 µM equivalent AQ concentrations for 72 h incubation. The IC_50_ values were determined using the point-point method. The detailed methods are provided in the Appendix A.

### 2.14. Clonogenic Assay

Clonogenic assay is an in vitro cell survival assay which is based on the single cancer cell’s capability to grow into a colony. A colony forming assay was performed to evaluate the long-term efficacy of AQ, AQ NP Rota, and FA-AQ NP toward colony inhibition. The protocol reported previously [39] was slightly modified and followed in this study. The detailed methods are provided in the Appendix A.

### 2.15. Three-Dimensional Spheroid Cell Culture Studies

An effective targeted cancer treatment is not only defined by enhanced cellular uptake or cytotoxic potential, but also by the enhanced penetrability of targeted nanoparticles into solid cancer cell masses. Three-dimensional spheroid cell culture studies are capable of simulating the in vivo characteristics of tumors, as reported in our previous studies [40]. The HeLa and MDAMB-231 cell lines were used in this assay. Briefly, 2000 cells/well were seeded in a Corning^®^ ultra-low attachment spheroid 96-well plate (Corning, NY, USA) and were incubated overnight at 37 °C/5% CO_2_ (Day 0). After 3 days of growth, all the spheroids were observed for spheroid formation with a rigid margin. The images were obtained using an inverted microscope at 10× magnification (Laxco, Mill Creek, WA, USA). The next day (Day 0 of treatment), half of the media was replaced with either the media (control) or the corresponding AQ, AQ NP Rota, or FA-AQ NP treatments (20 μM treatment used to maintain initial 10 μM treatment employed in the beginning), and images were obtained. In total, 100 µL of media was replaced with fresh media on further days of imaging. The images were captured on day 1, 3, 6, and 9, 12, and 15 days following treatment. ImageJ software (Version 1.44) was employed to calculate the diameters and the volumes of all spheroids.

### 2.16. Live-Dead Cell Assay

A live-dead cell assay was carried out on spheroids on day 15 of single dosing in the therapeutic model, as per the manufacturer’s procedure. The detailed methods are provided in the Appendix A.

### 2.17. Data Analysis and Statistical Evaluation

All data were referred to as the mean ± SD or mean ± SEM, with n = 3, unless mentioned otherwise. In total, 3 trials of cytotoxicity studies were performed for each control or treatment, with n = 6 for each trial data. The IC_50_ values were determined using the point-point method. An unpaired student’s *t*-test was used to compare the two groups and one-way ANOVA followed by Tukey’s multiple comparisons test was used to compare more than two groups using GraphPad Prism software (Version 7.04 for Windows, GraphPad Software, San Diego, CA, USA). A *p*-value of <0.05 was considered statistically significant and was presented in the data figures as a single asterisk (*). However, some studies demonstrated a smaller *p*-value of 0.01 or less, which is indicated at the respective places.

## 3. Results and Discussion

### 3.1. UPLC Method Development

A UPLC method was developed for AQ, as reported in a recent publication from our group [7]. The retention time was observed as 0.733 min with a total run time of 1.5 min, and a linearity range of 0.05 to 6 µg/mL.

### 3.2. Calibration Curve for FA

The calibration curve for FA was plotted between the UV absorbance and concentration of FA (Figure 1A). The linear regression data showed a linear relationship between 15.6 and 250 µg/mL for FA (r^2^ = 0.993). The linear regression equation was found to be as follows:Y = 0.001039X + 0.04648

### 3.3. Synthesis of PLGA-PEG-FA

PLGA-PEG-FA was synthesized by following a previously reported protocol [41]. The details are provided in the Appendix A.

### 3.4. Determination of Conjugation Efficiency

The PLGA-PEG-FA conjugate was analyzed for the FA-conjugated amount. The amount of conjugated FA was determined by comparing the obtained absorbance value with the standard curve. The amount of FA conjugated per 1 mg of PLGA-PEG-FA was found to be 27.8 ± 2.1 µg (Figure 2A). This observation confirmed the presence of FA on the nanoparticle surface.

### 3.5. Physicochemical Characterization: Particle Size, PDI, and Zeta Potential

Nanoparticle formulations, AQ NP Rota, and FA-AQ NPs were found to have average particle sizes of 197.5 ± 1.7 nm and 203.4 ± 8.8 nm, respectively. In addition, polydispersity indices (PDIs) of 0.2 or less suggested an overall uniform particle size distribution, as represented in Figure 2A. The evaluation of physicochemical characteristics of drug-loaded nanoparticles is important in drug delivery, as it significantly impacts the fate of encapsulated drugs in in vitro and in vivo conditions. Smaller particle sizes of targeted NPs with highly negative absolute zeta potentials enable higher accumulation in cancer cells [42]. The particle size of the nanoparticles strongly affect their ability to be internalized into cells in a tumor environment, i.e., the smaller the particle size, the higher the cellular internalization [42]. The negative zeta potential of both NPs was found to be ~−24 mV (Figure 2A). The high absolute zeta potential indicates a high electric charge on NPs’ surfaces, which can cause a strong repulsion among particles, so as to avoid the aggregation of the NPs.

### 3.6. Drug Content

AQ was efficiently loaded into the NPs, attaining % EE of 20.8 ± 1.6% (AQ NP Rota), 24.9.0 ± 5.4% (FA-AQ NP); and high % DL of 1.6 ± 0.1% (AQ NP Rota), 1.9 ± 0.4% (FA-AQ NP) (Figure 2A). As we utilized the pre-conjugated polymer (PLGA-PEG-FA) for nanoparticle fabrication instead of conjugating FA onto drug-loaded nanoparticles, the drug loss from surface-conjugated NPs during fabrication was minimal.

### 3.7. Morphological Studies

Representative TEM images of AQ NP Rota and FA-AQ NPs are presented in Figure 2B,C, revealing a spherical shape with a smooth surface for both nanoparticle formulations. No NP aggregation was observed during the TEM analysis, indicative of a relatively monodispersed size distribution of NPs in the formulation. The results were consistent with PDI and zeta potential values, as shown in Figure 2A.

### 3.8. Solid State Characterization

Powder X-ray Diffraction (PXRD): The AQ (Black) showed distinct peaks at 2θ values of 19.92° and 25.88°, while the AQ NP Rota (Red) and FA-AQ NP (Blue) demonstrated no AQ peaks, indicating an encapsulation of drug inside the nanoparticles (Figure 3A). These results were consistent with our earlier studies [7].

Differential Scanning Calorimetry (DSC): As seen in Figure 3B, thermogram of AQ showed a sharp endothermic peak at 166.8 °C, as reported in our previous studies [7,9]. The absence of a sharp peak in the AQ NP Rota and FA-AQ NP indicated extensive drug encapsulation within the NP core, and confirmed the XRD results, as described in an earlier section.

### 3.9. Stability Studies

Particle aggregation and emulsion instability is a significant concern for nanoformulations [35]. The stability analysis data revealed that FA-AQ NP demonstrated no significant changes in particle size, % entrapment, or zeta potential at 4 °C and 25 °C (Figure 4A–C), and, thus, was deemed stable.

### 3.10. Determination of Targeting Capability

Cellular Uptake Studies: The intracellular uptake of nanoparticles was quantified using A549 cells with 3 h of incubation (Figure 5A). In this study, AQ was substituted with coumarin (fluorescent) to quantify the uptake and accumulation. The fluorescent images captured following 3 h of incubation clearly demonstrated significantly higher internalizations compared to unencapsulated coumarin and non-targeted coumarin nanoparticles in A549 cells. In addition, the images demonstrated the higher accumulation of targeted nanoparticles around the nucleus (DAPI-stained), the preferred location for NP disruption and intracellular drug release [43]. FA on the FA-C NP enables NP interaction with cells, thus, further resulting in effective cellular internalization compared to that of C NP and plain coumarin.

Quantification of Cellular uptake by Determining Fluorescence Intensity: The cellular uptake of FA-C NP, C NP, and coumarin by the A549, HeLa, and MDAMB-231 cells was quantified by microplate reader-based fluorescence assays. The cells were able to internalize individual treatments to a varied extent, as shown in Figure 5B–D. At the 3-h time point, the FA-C NP uptake was ~7.6, 2.6 (A549); ~2.3, 1.6 (HeLa); and ~5.7, 1.8 (MDAMB-231) times higher compared to plain coumarin and C NP, respectively. The fluorescence intensity of FA-C NP (23,230.7 ± 8927.8: A549; 37,276.0 ± 8773.1: HeLa; 42,113.3 ± 6115.8: MDAMB-231) treated cells was significantly higher when compared with C NP (9042.6 ± 5819.8: A549; 23,151.0 ± 7241.0: HeLa; 23,830.0 ± 4587.6: MDAMB-231) and coumarin (2991.3 ± 127.8: A549; 16,300.0 ± 3576.3: HeLa; 7434.0 ± 4431.6: MDAMB-231) at the 3-h time point (A549: coumarin vs. FA-C NP: *p* < 0.05; HeLa: coumarin vs. FA-C NP: *p* < 0.05; MDAMB-231: coumarin vs. FA-C NP: *p* < 0.05). The quantitative results indicate congruency to the fluorescence imaging, as shown in Figure 5A. Due to the presence of FA on the nanoparticle surface, the uptake of the FA-C NP into cells is enabled by the interaction between FA and the cell surface-expressed folate receptors, resulting in an elevated internalization in comparison to C NP and coumarin (A549) or plain coumarin (Hela, MDAMB-231). The presence of FA on NPs’ surface is associated with better uptake and significantly higher cytotoxicity in cancer cells, explaining the specificity of FA-mediated binding of conjugated NPs [44]. In addition, the anticipated process for the internalization of negatively charged nanoparticles could be endocytosis [42]. Recently, research work conducted by Angelopoulou et al. has demonstrated the superior ability of FA-conjugated pegylated magnetic NPs to deliver doxorubicin to cancer cells of solid tumors [45].

### 3.11. Cytotoxicity Studies

As seen in Figure 6, it can be seen that AQ cytotoxicity against the A459, HeLa, and MDAMB-231 cell lines was significantly enhanced by nanoparticle encapsulation. Figure 6A–C illustrates the cytotoxic effects of AQ, AQ NP Rota, and FA-AQ NP in the MDAMB-231, HeLa, and A549 cell lines, respectively. While the HeLa and MDAMB-231 cell lines possess well-characterized folate receptor expression, the A549 cells demonstrate a relatively moderate expression of folate receptors [19,46,47]. Nasiri et al. employed the HeLa and MDAMB-231 cell lines to perform in vitro experiments for assessing the anti-cancer efficacy of FA-targeted dual mode nanoparticles to achieve the targeted delivery of bromelain [48].

The IC_50_ values for AQ, AQ NP Rota, and FA-AQ NPs were found to be 18.3 ± 4.5 µM, 11.5 ± 11.8 µM, and 12.7 ± 9.8 µM in the HeLa cell line (Figure 6B); and 17.5 ± 6.6 µM, 30.0 ± 17.5 µM, and 11.5 ± 8.3 µM in the MDAMB-231 cell line, respectively (Figure 6A). All IC_50_ values are tabulated in Figure 6E. The smaller and monodispersed particle size and the targeting capability of FA-AQ NPs facilitate the efficient internalization and higher accumulation of nanoparticles inside the cells, thus achieving a higher cytotoxicity at the same dosing amount. IC_50_ of FA-AQ NP was calculated to be significantly lower than that of the free drug and also non-targeted NPs without FA-conjugation in the case of MDAMB-231 cells with documented FA receptor overexpression, in comparison to A549 (AQ NP Rota vs. FA-C NP: *p* < 0.0001) [18]. No significant difference in the IC_50_ values of plain AQ and FA-targeted NPs was observed in the A549 cells (Figure 6C). In vitro cytotoxicity studies of Blank NP and Blank FA-NP were also performed on the HEK-293 cell line. Both Blank NP and Blank FA-NP were found be safe from % cell viability determinations after incubating the HEK cell line for 72 h, which has been represented in Figure 6D. In none of the cases was the cell viability below 70% using blank nanoparticles equivalent to concentrations ranging from 0.39 to 50 µM, suggesting that the formulation components were not toxic by themselves. Wadhawan et al. conjugated the folate ligand with the PLA–PEG copolymer to obtain the active targeting of the system in the MDAMB-231 cancer cells. Folic acid-conjugated nanoparticles were reported to actively target the cells and deliver therapeutic agent to the target site more efficiently [49].

### 3.12. Clonogenic Assay

AQ, AQ NP Rota, and FA-AQ NP were tested for their long-term efficacy by performing clonogenic assay in HeLa and MDAMB-231. From Figure 7A, it can be illustrated that clonal expansion was significantly inhibited by FA-AQ NP compared to AQ and AQ NP Rota in both the HeLa and MDAMB-231 cell lines. After 48 h of the treatment period and 7 days of incubation with 10 µM AQ, AQ NP Rota, and FA-AQ NP, the percentages of colonies that survived in the HeLa cells were 20.9 ± 5.3%, 29.9 ± 6.6%, and 9.2 ± 1.1%, respectively (Figure 7B, AQ NP Rota vs. FA-AQ NP: *p* < 0.05, control vs. FA-AQ NP: *p* < 0.05), considering the number of colonies to be 100% in the drug-free treatment control wells. In the case of the MDAMB-231 cell line, the percentage of colony growth was found to be 57.6 ± 1.8%, 84.8 ± 9.2%, and 43.2 ± 9.9% with AQ, AQ NP Rota, and FA-AQ NP (Figure 7C; AQ NP Rota vs. FA-AQ NP: *p* < 0.05, control vs. FA-AQ NP: *p* < 0.05) The data suggest an approximately 3.2-fold (HeLa) and 2-fold (MDAMB-231) greater reduction in colony-forming with FA-AQ NP as compared to AQ NP Rota. The reason for this could be due to the higher internalization of AQ through the FA receptors on the surface of the HeLa and MDAMB-231 cells, leading to higher internalization of AQ into these cells. Thus, FA-conjugated AQ NPs could have established a higher cytotoxicity and consequently lowered colony-forming ability, as explained by Nasiri et al. [48], where the authors revealed the higher clonogenic inhibitory effects of FA-conjugated NPs on MDA-MB-231 compared to the plain drug [48]. In another study, Thulasidasan et al. reported the superior ability of curcumin-entrapped in PLGA-PEG nanoparticles conjugated to folic acid to inhibit the colony formation of HeLa cells compared to free curcumin [50]. These data represent the FA-AQ NP’s efficacy in curbing the metastatic or new tumor growth possibility by reproduction inhibition, and can clearly be linked to improve the intracellular (and intratumoral) accumulation of the drug with nanoparticles’ formulation [48].

### 3.13. Three-Dimensional Spheroid Studies

A two-dimensional cell culture experiment does not accurately mimic solid tumor structure, drug resistance, and drugs’ poor penetrability due to the lack of tumor microenvironment [51]. Furthermore, 3D-cultured spheroids combine the ability to mimic the 3D structure of malignant tissues precisely, in addition to being relevant to cancer microenvironment to which the tumor cells are exposed [52,53].

For this spheroid study, we chose the HeLa and MDAMB-231 cell lines. This study was performed using a single dose regimen with drug treatment starting 3 days after plating the cells for spheroid formation. All drug treatments were conducted at 10 µM AQ concentration. The spheroid images were captured using an inverted microscope (Laxco, Mill Creek, WA, USA) and were analyzed using the ImageJ (Version 1.44; National Institute of Health, Bethesda, MD, USA). The representative spheroid images (HeLa: Figure 8A, MDAMB-231: Figure 8B), and calculated spheroid volume comparisons (Figure 9A,B) are presented. For both the HeLa and MDAMB-231 cell lines, control spheroids were found to be growing in size/volume over 15 days, whereas the growth of nanoparticle-treated spheroids was significantly inhibited (Figure 8A,B).

In this single dose study with a single treatment given on day 1, the control spheroids of the HeLa cell line were found to have volumes of 64.1 ± 2.7 mm^3^ on day 15, as compared to 5.2 ± 0.2 mm^3^ volume on day 1, a 12.3-fold increase in the tumor volume. Following the 15-day treatment, a significant reduction in spheroid volumes was observed for FA-AQ NP (39.3 ± 2.3 mm^3^), compared to 51.8 ± 3.8 mm^3^ (plain AQ) and 53.3 ± 7.0 mm^3^ (AQ NP Rota) (control vs. FA-AQ NP *p* < 0.0001; AQ vs. FA-AQ NP *p* < 0.05) (Figure 9A).

While MDAMB-231 control spheroids were found to have volumes of 11.4 ± 0.7 mm^3^ on day 15, as compared to 6.7 ± 0.5 mm^3^ volume on day 1, a 1.7-fold increase in the tumor volume. Following the 15-day treatment, a significant reduction in spheroid volumes was observed for FA-AQ NP (8.8 ± 1.6 mm^3^) compared to 14.2 ± 1.6 mm^3^ (plain AQ);and 13.0 ± 1.3 mm^3^ (AQ NP Rota) (AQ vs. FA-AQ NP *p* < 0.05) (Figure 9B). These results underlined a more-pronounced tumor reduction potential of FA-AQ NP, due to the enhanced cellular interaction and tumor penetration capabilities of the targeted nanoparticles. Treatment with FA-AQ NP resulted in a significant difference in spheroid volumes as compared to AQ- and control-treated groups in both cell lines. Tumor growth suppression was continually observed until the end of the treatment. It can be understood that, through the addition of a ligand moiety (FA), on NPs’ surface, NPs are directed against receptors (FA) expressed at the cell surface, concurrently enhanced uptake and the distribution of NPs in the tumor mass even with complicated tumor microenvironment [54].

Wang et al. evaluated FA-modified luteolin-loaded nanoparticles in breast cancer treatment. The results from the in vitro studies revealed the excellent penetration abilities of FA-modified NPs in the tumor spheroids of breast cancer cells, thus facilitating targeted drug delivery to the deep regions of the tumor. The authors also evaluated the NPs’ efficacy in vivo and found that FA modification enhanced the anti-tumor activity of Luteolin-loaded NPs. These results were in accordance with the in vitro anti-tumor efficacy results [55]. In another study, Sriraman et al. evaluated different liposomal formulations for their targeting ability and anti-tumor efficacy in vitro, spheroids, and in vivo. Furthermore, the authors investigated the correlation among the three experimental models after treating with different targeted groups. The dual-targeted liposomes showed a significant cytotoxicity compared to the untargeted formulations, both in vitro and in vivo [47]. Three-dimensional spheroid studies have been utilized in our earlier studies to evaluate the anti-cancer efficacy of transferrin-conjugated AQ-loaded nanoparticles. The results revealed that transferrin-conjugated NPs had superior tumor penetrability and, hence, exhibited tumor reduction potential [9]. In another study from our group, breast cancer spheroids of MDAMB-231 were utilized to determine the effectiveness of Metformin-loaded liposomes. A tumor growth suppression was exhibited in the case of liposomal formulations [8].

### 3.14. Live-Dead Cell Assay

Defining viable and dead cells within a solid spheroid mass can be carried out by live-dead cell assay. Developed drug-loaded nanocarriers enable drug’s entry into the inner core of the tumor while 2D tumor volume measurements may not reveal a nanoformulation’s efficacy in penetrating the tumor core, as the spheroid masses consist of dead cell cores surrounding the structural periphery [7,56]. Three-dimensional spheroids are also grown over an extended period, with intratumoral core zones being exposed to environmental conditions, such as lack of O_2_, and nutrients, developing necrotic and quiescent cells in the core regions [56]. Therefore, it is essential to quantify actual dead cell portions out of a spheroid mass. Figure 9C,D represents terminal spheroid images on day 15 after treatment and the quantification plot of red fluorescence intensity, respectively. While AQ NP Rota and FA-AQ NP treatments demonstrated a reduction in the tumor volume, higher red fluorescence (dead cells) was seen with the FA-AQ NP treatment in the case of HeLa spheroids, as seen in Figure 9D. The ratio of red fluorescence protein (RFP) to green fluorescence protein (GFP) comparison on day 15 of the single dose (FA-AQ NP) post-treatment period revealed that RFP/GFP is 1.5-, 1.4-, and 1.3-folds higher in comparison to the control, AQ, and AQ NP Rota (control vs. FA-AQ NP: *p* < 0.05, control vs. AQ NP Rota: *p* < 0.01). These results indicate a higher red fluorescence comparatively in the case of FA-AQ NP treatment, which could be due to the targeting potential of FA-conjugated NPs to penetrate spheroids and exhibit their anti-cancer efficacy. Figure 9E represents the plot for RFP/GFP against AQ, AQ NP Rota, and FA-AQ NP treatments in MDAMB-231 cells.

## 4. Conclusions

With the results demonstrated in the current study, it can be concluded that FA-conjugated nanoparticles of AQ were successfully formulated using a scalable HPH method. Our findings from the physicochemical characterization of developed NPs suggest that they have optimal particle sizes and surface charges enabling effective cellular uptake. Developed FA-conjugated AQ NPs were compared with their counterpart (un-conjugated AQ NPs) in their anti-cancer efficiency against various cancer cell lines. While supporting this idea, extensive in vitro studies, such as cytotoxicity studies and 3D spheroid studies, and colony growth inhibition assays were carried out where superior efficacy of FA-conjugated NPs had been confirmed. Specifically, the FA-conjugated NPs exhibited an anti-cancer efficacy in the MDAMB-231 cell line predominantly where FA receptors were over-expressed compared to the A549 or HeLa cell lines. This demonstrates the importance of FA conjugation in targeting FA receptors on cancer lines, resulting in an effective treatment. In future investigations, we would carry out preclinical pharmacokinetic and pharmacodynamic studies for the further evaluation of amodiaquine-loaded folic acid-conjugated nanoparticles for assessing the true targeting capability of developed nanoparticles.

## Figures and Tables

**Figure 1 pharmaceutics-15-01001-f001:**
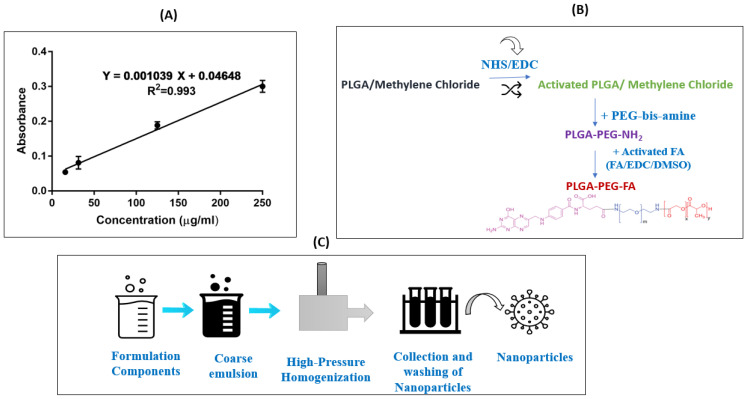
(**A**) Calibration curve for FA. (**B**) Schematics of PLGA conjugation with folic acid. (**C**) Graphical representation of nanoparticles’ preparation.

**Figure 2 pharmaceutics-15-01001-f002:**
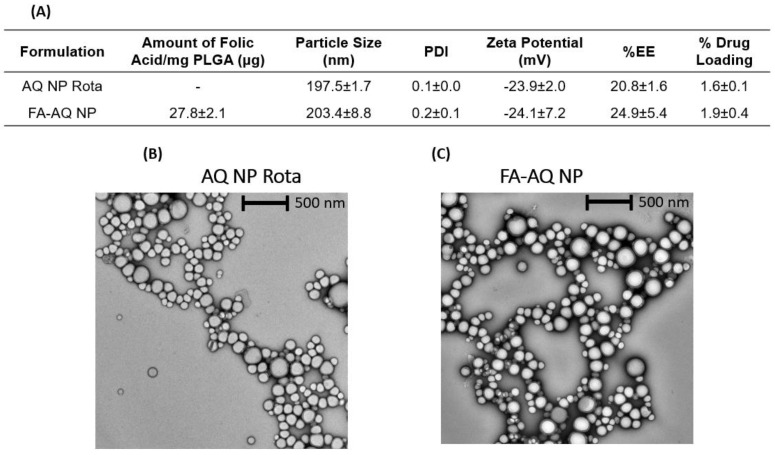
(**A**) Characterization of non-targeted and targeted nanoparticles. (**B**,**C**) Transmission electron microscopy (TEM) images of AQ NP Rota and FA-AQ NPs. The scale bar represents 500 nm. Magnification 20 k×.

**Figure 3 pharmaceutics-15-01001-f003:**
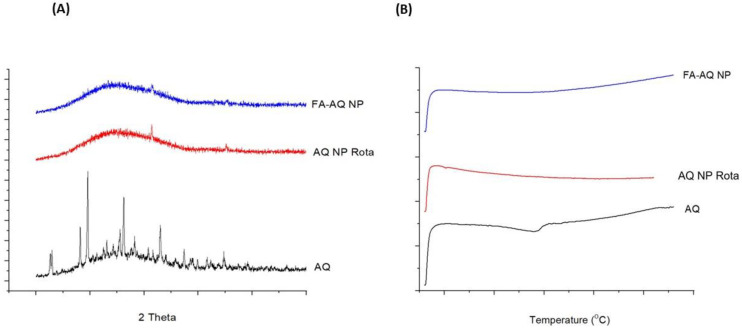
(**A**,**B**) Solid state characterization studies: (**A**) XRD patterns of AQ, AQ NP Rota, and FA-AQ NP; (**B**) Thermograms of AQ, AQ NP Rota, and FA-AQ NP.

**Figure 4 pharmaceutics-15-01001-f004:**
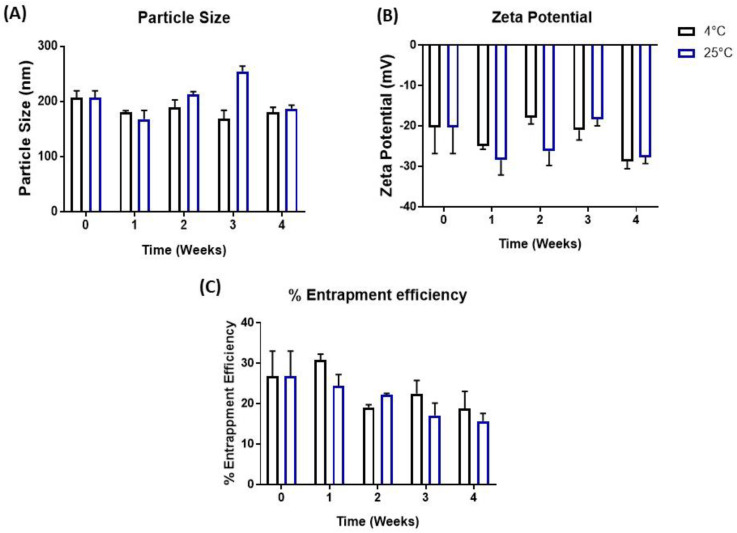
Influence of storage temperature and length of storage on (**A**) particle size, (**B**) zeta potential, and (**C**) % drug entrapment efficiency of FA-AQ NPs. Formulations were stored at 4 °C and 25 °C over a period of 4 weeks. Data represent mean ± SD (n = 3).

**Figure 5 pharmaceutics-15-01001-f005:**
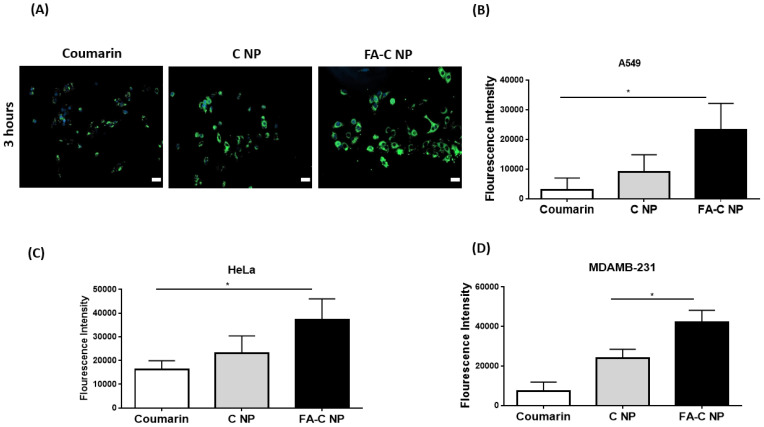
Intracellular uptake of coumarin-6-loaded nanoparticles (plain and FA-targeted). (**A**) Fluorescence microscopy images in A549 at 3 h. Nuclei are stained blue (DAPI), and C NP and FA-C-NP are green. Scale bar = 100 μm. Representative images from n = 3 experiments. Scale bar = 100 µm. (**B**–**D**) Quantitative representation of cellular uptake through fluorescence measurements in A549 (**B**), HeLa (**C**), and MDAMB-231 (**D**) cell lines. Data represent mean ± SD (n = 3). ** p* < 0.05.

**Figure 6 pharmaceutics-15-01001-f006:**
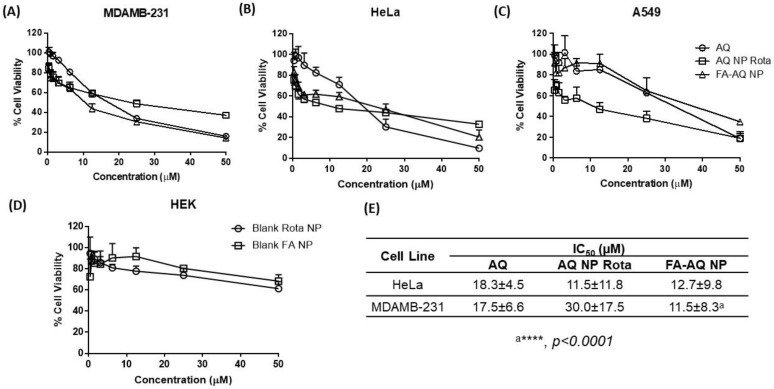
Inhibitory effects on different cell lines (**A**) MDAMB-231, (**B**) HeLa, (**C**) A549 after treatments with AQ, AQ NP Rota, and FA-AQ NP. (**D**) Cell viability in HEK-293 cell line after treatment with blank non-targeted and FA-targeted NPs; and (**E**). IC_50_ of AQ, AQ NP Rota, and FA-AQ NP in HeLa and MDAMB-231. Data represent mean ± SD (n = 6) of at least 3 independent trials. (**A**–**C**). Data represent mean ± SD (n = 6) (**D**). ***** p* < 0.0001.

**Figure 7 pharmaceutics-15-01001-f007:**
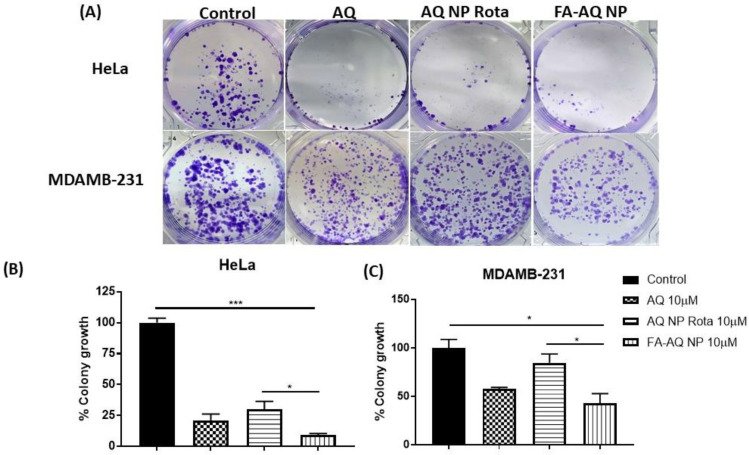
(**A**) Representative images showing distinct colonies after staining in HeLa and MDAMB-231 cell lines. Three different experiments were performed. (**B**,**C**) Quantitative representation of clonogenic assay as % colony growth with AQ, AQ NP Rota, and FA-AQ NPs treatment as compared to control in HeLa and MDAMB-231 cell lines, respectively. Data represent mean ± SEM (n = 3). ** p* < 0.05 and *** *p* < 0.001.

**Figure 8 pharmaceutics-15-01001-f008:**
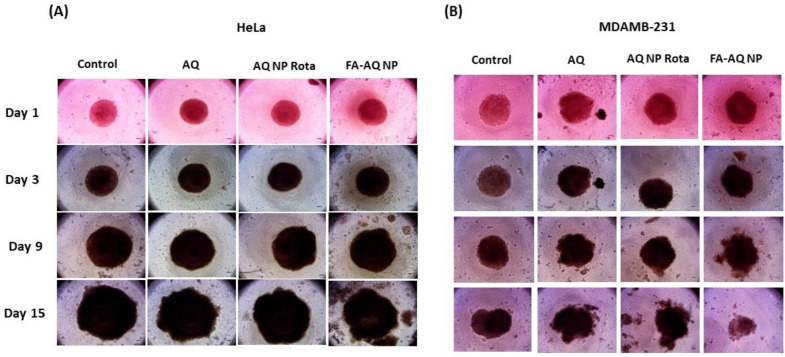
Three-dimensional spheroid study: (**A**,**B**) Spheroid images of HeLa (**A**) and MDAMB-231 (**B**) cell lines after treating with control, AQ, AQ NP Rota, and FA-AQ NP. Images represent n = 8 for each treatment. Images were captured using Evos-FL fluorescence microscope with 4× objective. Scale bar is 400 µm.

**Figure 9 pharmaceutics-15-01001-f009:**
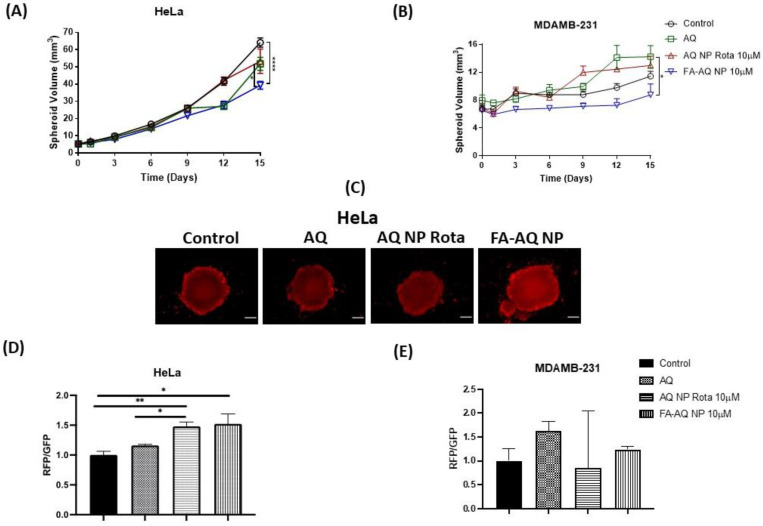
Three-dimensional spheroid study: Spheroid volume comparison plots for HeLa (**A**) and MDAMB-231 (**B**) spheroids. Data represent mean ± SEM (n = 8). *******
*p* < 0.05 and ********
*p* < 0.0001. (**C**) Live-dead cell assay: Red fluorescence intensity demonstrated by HeLa spheroids. Images were captured using Evos-FL fluorescence microscope with 4× objective. (**D**) The graph represents comparison of red fluorescence intensity/green fluorescence intensity of HeLa spheroids. (**E**) The graph represents comparison of red fluorescence intensity/green fluorescence intensity of MDAMB-231 spheroids Data represent mean ± SEM (n = 4). *****
*p* < 0.05, ******
*p* < 0.01.

## Data Availability

Data available upon request from corresponding author.

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
