# Peer review of "Development and Characterization of Folic Acid-Conjugated Amodiaquine-Loaded Nanoparticles–Efficacy in Cancer Treatment"

_pharmaceutics, 2023, doi:10.3390/pharmaceutics15031001_

Round 1

Reviewer 1 Report

In this study, folic acid (FA) was conjugated with PLGA polymer followed by formulation of AQ- loaded NPS. Developed folic acid conjugated nanoparticles demonstrated uniform particle size distribution, had a visible spherical shape under transmission electron microscopy. The anticancer activity of FA conjugated amodiaquine loaded nanoparticles (FA-AQ NP) against FA-receptors expressing cancer cells was evaluated. However, this work need major improvement before it can be considered for publication: 1. What was the unique advantage of PLGA-PEG-FA conjugation for drug dilivery compared to other conjugation systems? 2. The drug loading efficiency is below 2%, is this good enough compare to others reported in literature  ? 3. Most of the figures throughout the manuscript are poorly organized and reader-unfriendly. For example, the chemical structure of PLGA-PEG-FA is ugly. 4. There should be a space between the numbers and the following units. 5. In Figure 8, "400µM" should be "400 µm". 6. Why only choose AQ as a loading drug, how about the loading efficiency and therapeutic performance of other common chemo-drugs? 

Author Response

RESPONSE TO REVIEWER’S COMMENTS (Reviewer #1)

In this study, folic acid (FA) was conjugated with PLGA polymer followed by formulation of AQ- loaded NPS. Developed folic acid conjugated nanoparticles demonstrated uniform particle size distribution, had a visible spherical shape under transmission electron microscopy. The anticancer activity of FA conjugated amodiaquine loaded nanoparticles (FA-AQ NP) against FA-receptors expressing cancer cells was evaluated. However, this work needs major improvement before it can be considered for publication: 

Author Response: Thank you for your review and valuable comments. We have provided response to all your critiques in the section below.

Comments:

  1. What was the unique advantage of PLGA-PEG-FA conjugation for drug delivery compared to other conjugation systems? 

Author Response: Thank you for your valuable comment. PLGA-PEG-FA conjugated functionalized polymers have been utilized in several studies due to its superior targeting characteristics [1,2]. Similarly, inclusion of a polyethylene glycol (PEG) spacer is a well-established strategy to increase hydrophilicity and avoid aggregation. In addition, PEGylation provides passive tumor-targeting via enhanced permeability and retention effect. These delivery systems are recognized selectively by the folic acid receptors in tumor cells to acquire receptor-mediated active tumor-targeting [3]. These details were added in the manuscript (Lines 80-86).

  1. The drug loading efficiency is below 2%, is this good enough compared to others reported in literature? 

Author Response: Thank you for your valuable comment. Based on the efficacy of FA-AQ NPs at lower concentrations, a lower drug loading efficiency would be sufficient to exhibit anti-cancer effects. Approximately similar drug loading efficiencies were observed in our previous research studies, which demonstrated superior anti-cancer effects [4,5]. In addition, utilization of targeted drug delivery systems facilitates superior anti-cancer efficacies at the target site, thus further reducing the required doses.

  1. Most of the figures throughout the manuscript are poorly organized and reader-unfriendly. For example, the chemical structure of PLGA-PEG-FA is ugly. 

Author Response: Thank you for your comment. The chemical structure of PLGA-PEG-FA has been updated for a clear representation. Fig. 1 has been updated with graphical presentation of nanoparticles’ preparation process. Figures have been reorganized for clear presentation and updated Figure 1 has been submitted.

  1. There should be a space between the numbers and the following units. 

Author Response: Thank you for your comment. Space has been removed between numbers and the following units throughout the manuscript.

  1. In Figure 8, "400µM" should be "400 µm". 

Author Response: Thank you for your comment. "400µM" has been replaced with "400 µm" in Figure 8.

  1. Why only choose AQ as a loading drug, how about the loading efficiency and therapeutic performance of other common chemo-drugs? 

Author Response: Thank you for your valuable comment. This study is focused on investigating amodiaquine as a potential cancer therapeutic by targeting its delivery to cancer cells and tissues. Amodiaquine has been explored for its anti-cancer efficacy in different cancer types by our research group. Our group has been a pioneer in establishing AQ and other antimalarials as viable poly-pharmacological therapeutic moieties for various diseases with autophagic flux being one of the contributors. We have published several studies about Repurposing potential of amodiaquine, both as plain drug, and also drug-loaded nanoparticles in treating non-small cell lung cancer and breast cancer [4,6,7]. In the current study, potential of amodiaquine has been further explored for its anti-cancer efficacy using folic acid conjugation while focusing on targeted drug delivery. While it would be valuable to test other chemo drugs as well, it will not fit within the scope of current study, which focuses on folic acid-assisted targeted delivery of amodiaquine. Our research group is routinely developing new delivery systems where we test multiple new and approved chemotherapeutic agents.

Author Response

RESPONSE TO REVIEWER’S COMMENTS (Reviewer #2)

This study was to construct amodiaquine-loaded, folic acid-conjugated polymeric nanoparticles (FA-AQ NP) to treat cancer, that might be scaled to commercial production. Several in-vitro cell culture studies including cytotoxicity studies and 3D spheroid studies, assays exhibiting colony formation inhibitory behavior where superior efficacy of FA conjugated NPs has been confirmed. In addition, some issue concerned:

Comments:

  1. One of the research contents of this subject is the preparation of targeted preparations, but there is a lack of discussion on the preparation process and process parameters, including the selection of carriers, the influence of various factors of preparation process on the quality of preparations. The reference documents of the preparation process mentioned in the article, the connected target and the route of drug delivery are different from those in this article.

Author Response: Thank you for your comment. An established folic acid conjugation method previously reported by Fasehee et al. [8] has been followed in the current study with slight modifications. Respective references have been corrected and cited in the updated manuscript.

  1. Only a brief synthetic scheme is presented, but the identified structure of the product is missing. And the result of successful conjugated drugs, target and carrier materials cannot be obtained. Please supplement the data of the successful conjugation by IR or NMR spectroscopy.

Author Response: Thank you for your valuable comments. Folic acid conjugation method reported in a previous publication [8] has been utilized in the current study. The cited study reported successful conjugation of folic acid under respective conditions and process parameters. Hence, additional studies were not conducted as per the scope of the current study.

  1. For the UPLC method, the representative picture should provide the blank, standard and sample chromatogram that can reflect the specificity. The supplementary materials only provided a chromatogram without specific sample.

Author Response: Thank you for your valuable comment. A representative UPLC chromatogram was provided in the Supplementary materials for reference purposes only. Sample and standards resulted in similar characteristic peaks and retention time for their quantification. To avoid confusion, reference chromatogram has now been removed from the Supplementary information.

  1. The absorption and distribution environment of drugs in vivo is complex, and only in vitro cell experiment cannot reflect the true targeting capability and anticancer activity. Why there is no research data on targeting capability and anticancer activity in vivo?

Author Response: Thank you for your valuable comments. We agree with the reviewer’s comment on in-vivo study. However, current work involves in-vitro evaluation of amodiaquine’s anticancer efficacy in 2D and in physiologically relevant 3D models. In addition, recent guidelines from the FDA encourage use of non-animal models to evaluate efficacy of new and investigational drugs [9]. As 3D in-vitro models allow creation of tumoral mass in three dimensions while mimicking the in-vivo environment and tumor growth, they can recapitulate in-vivo biology. Moreover, these models promote aggregation of cells which assists in constructing an in-vitro model with infrastructure of in-vivo tumor formation. Although preclinical in-vivo experiments were not performed for this study, many recent studies have compared in-vitro 3D spheroids with in-vivo efficacy and have demonstrated to have similar outcomes in both the models [10,11]. In addition, spheroid culture demonstrates metabolic similarities to the original tissue and in-vivo experiments; and is being extensively used to replace animal experiments, not just for cancer but also for other diseases [12]. In future investigations, we would carry out preclinical pharmacokinetic and pharmacodynamic studies for further evaluation of amodiaquine loaded folic acid conjugated nanoparticles. These details have now been included in the conclusion. (Lines 592-594)

  1. Please check the expression of line 17 “Futhermore, cytotoxicity studies showed the superior efficacy of FA-AQ NP in different cancer cells such as”,and line 504 “(AQ NP Rota Control vs (AQ vs FA-AQ NP p<0.05)”

Author Response: Thank you for your comment. In line 17 “Futhermore, cytotoxicity studies showed the superior efficacy of FA-AQ NP in different cancer cells such as” has been updated to “Furthermore, cytotoxicity studies showed the superior efficacy of FA-AQ NP in different cancer cells such as MDAMB-231 and HeLA”. In line 516 “(AQ NP Rota Control vs (AQ vs FA-AQ NP p<0.05)” has been updated to “(AQ NP Rota) (AQ vs FA-AQ NP p<0.05)”

  1. Why there is no experimental data of in vitro release?

Author Response: Thank you for your comment. In our previously published work on amodiaquine nano-formulations, in-vitro release studies were performed on amodiaquine nanoparticles (un-conjugated, and transferrin conjugated). Amodiaquine loaded nanoparticles were evaluated for their ability to release drug in pH 7.4 buffer and results have indicated complete drug release within 8 hours [6]. Furthermore, transferrin conjugated were found to exhibit >80% drug release in phosphate buffer pH 5.5 (simulating tumor environmental pH) within first 4 hours, indicating the drug release from the formulations after their internalization into tumor cells [4]. In both the studies, efficient drug release had been demonstrated from developed nanoparticles. Hence, no in-vitro release studies have been carried out in the current study.

Reviewer 3 Report

1. Update Fig. 1. with a graphical representation of bioconjugation and nanoparticle production, instead of the flow chart.

2. Author's individual contribution must be defined at the end of the manuscript. Please update.

3. Institutional Review Board Statement and Informed consent statement, either can be omitted or addressed properly.

4. More than 12 self-cited references are in the m/s. Kindly remove.

Author Response

RESPONSE TO REVIEWER’S COMMENTS (Reviewer #3)

Comments:

  1. Update Fig. 1. with a graphical representation of bioconjugation and nanoparticle production, instead of the flow chart.

Author Response: Thank you for your comment. Figure 1 has been updated with a clear presentation of nanoparticles’ preparation process. Updated Figure 1 has been submitted.

  1. Author's individual contribution must be defined at the end of the manuscript. Please update.

Author Response: Thank you for your comment. We provided individual author contribution during article submission process. For further clarity, individual contributions for all authors have been defined at the end of the manuscript as well.

  1. Institutional Review Board Statement and Informed consent statement, either can be omitted or addressed properly.

Author Response: Thank you for your comment. Institutional Review Board Statement and Informed consent statement have been removed from the manuscript.

  1. More than 12 self-cited references are in the m/s. Kindly remove.

Author Response: Thank you for your valuable comments. Several self-cited references have been removed from the manuscript.

Reviewer 4 Report

This manuscript investigated the preparation of FA-modified Amodiaquine-Loaded PLGA nanoparticles and evaluated the targeting antitumor capacity in different tumor cell lines and tumor Spheroid. The fabrication and physicochemical characterizations were introduced carefully, however, there are some issues of biological experiments required improvement and explanation.

(1)   In the Introduction, some contents talking about FA receptor in the 3rd paragraph and 5th paragraph were repeated. 

(2)   Figure 1B was not mentioned in the main text.

(3)   In Figure 5, the cellular nucleus should be stained to clearly show the cells and the internalized NPs. Moreover, images with a higher magnitude should be presented to allow readers see the cells clearly.

(4)   In Figure 7, why the cell numbers for Hela and MDAMB231 were set significantly different in the control groups? Did it mean the Hela cells were not sensitive to the FA-modified drug delivery system?

(5)   In Figure 8A, it was hardly to tell the inhibitory effects of the tested drugs, no matter AQ, AQ-NP Rota, and FA-AQ-NP. Again, it seemed the Hela cells were not sensitive to the FA-modified drug delivery system. Please provide more explanation.

(6)   In Figure 9C, were there any living cells in each group? Why there were not any green fluorescence observed? Please provide more explanation.

(7)   Although A549, Hela and MDAMB231 were used in this study, it seemed that the expected antitumor activity for FA-AQ-NP was confirmed in the MDAMB231, the breast cancer tumor cells, while Hela cells seemed not able to respond FA-AQ-NP well, and data for A549 were not sufficient to demonstrate the effectiveness of FA-AQ-NP.  Therefore, the Conclusion should be changed, more specific conclusion should be given.

Author Response

RESPONSE TO REVIEWER’S COMMENTS (Reviewer #4)

This manuscript investigated the preparation of FA-modified Amodiaquine-Loaded PLGA nanoparticles and evaluated the targeting antitumor capacity in different tumor cell lines and tumor Spheroid. The fabrication and physicochemical characterizations were introduced carefully, however, there are some issues of biological experiments that required improvement and explanation.

Author Response: Thank you for your comments. We have now modified the manuscript as per your comments.

Comments:

  1. In the Introduction, some contents talking about FA receptor in the 3rdparagraph and 5th paragraph were repeated. 

Author Response: Thank you for your comment. The repeated sentence has been removed from 5th paragraph of the Introduction. (Lines 71-72)

  1. Figure 1B was not mentioned in the main text.

Author Response: Thank you for your comment. Figure 1B has been mentioned in Section 2.4, Line 141.

  1. In 5, the cellular nucleus should be stained to clearly show the cells and the internalized NPs. Moreover, images with a higher magnitude should be presented to allow readers to see the cells clearly and updated figure have been submitted.

Author Response: Thank you for your comment. All the microscopic images in Fig. 5 have been reanalyzed and uniformly adjusted to clearly exhibit the DAPI-stained nuclei and cells. Updated Fig. 5 has been submitted.

  1. In Figure 7, why the cell numbers for Hela and MDAMB231 were set significantly different in the control groups? Did it mean the Hela cells were not sensitive to the FA-modified drug delivery system?

Author Response: Thank you for your comment. Clonogenic assay presented in Fig. 7 was performed using same protocol and same number of cells for both HeLa and MDAMB-231 cell lines. The difference in the number of colonies observed in Fig. 7 refers to the varying abilities of HeLA and MDAMB-231 to proliferate (doubling time for HeLA is about 34 hours, compared to about 24 hours for MDAMB-231), depending on the characteristics of each cell line; and not different treatment conditions/cell numbers.

  1. In Figure 8A, it was hardly to tell the inhibitory effects of the tested drugs, no matter AQ, AQ-NP Rota, and FA-AQ-NP. Again, it seemed the Hela cells were not sensitive to the FA-modified drug delivery system. Please provide more explanation.

Author Response: Thank you for your comment. Fig. 8A refers to the representative spheroid images from n=6 per each time point for each treatment group. These images merely provide qualitative determination on the effect of treatments on spheroid size. Fig. 9A represents the quantitative evaluation of spheroid volumes considering n=6 spheroids for each treatment type, where a clear and significant difference can be seen among different treatment groups vs. control.

  1. In Figure 9C, were there any living cells in each group? Why there were not any green fluorescence observed? Please provide more explanation.

Author Response: Thank you for your valuable comment. Fig. 9C represents the specific images captured for red fluorescence (dead cells) only. These images have been included to better represent the significance of treatments towards cancer cell death, as demonstrated by increasing number of dead cells with targeted nanoparticle treatment.

  1. Although A549, Hela and MDAMB231 were used in this study, it seemed that the expected antitumor activity for FA-AQ-NP was confirmed in the MDAMB231, the breast cancer tumor cells, while Hela cells seemed not able to respond FA-AQ-NP well, and data for A549 were not sufficient to demonstrate the effectiveness of FA-AQ-NP.  Therefore, the Conclusion should be changed, more specific conclusion should be given.

Author Response: Thank you for your valuable comment. We agree with the reviewer’s comment on conclusion. Amodiaquine was found to demonstrate anti-cancer efficacy predominantly in MDAMB-231 cell line, as reported in our recent publication [7]. Conclusion has now been updated, to reflect the observations. Specifically, FA conjugated NPs exhibited anti-cancer efficacy in MDAMB-231 cell line predominantly where FA receptors are over-expressed compared to A549 or HeLa cell lines. This demonstrates the importance of FA conjugation in targeting FA receptors on cancer lines resulting in an effective treatment. (Lines 588-594)

Reviewer 5 Report

The study developed folic acid-conjugated polymeric nanoparticles loaded with amodiaquine (FA-AQ NP) for cancer treatment. The nanoparticles demonstrated uniform size distribution and increased cellular uptake in various cancer cell types. The FA-AQ NP exhibited superior efficacy and antitumor ability compared to other treatments, making it a promising drug delivery system for cancer therapy.

Couple of comments for the authors:

When reading the introduction, there are a couple of questions that come to my mind that should be answered or referred within the text, including references:

1.     What is the mechanism of action of amodiaquine that makes it effective against cancer?

2.     What are the limitations of the current chemotherapy regimen besides resistance, adverse events/toxicity, and poor targetability?

3.     What other nanocarrier systems, besides liposomes and polymeric nanoparticles, are developed for selective targeting against tumors?

4.     What is the reason for the restricted expression of folate receptors in normal tissues?

5.     How does the presence of high receptor affinity of FA result in high tumor tissue specificity?

6.     What are the features of FA that make it a promising targeting ligand?

7.     How do folate receptors play a crucial role in iron homeostasis?

8.     What are the potential advantages of using FA conjugates for targeting FRα receptors?

9.     What is the evidence that targeting FA receptors through FA offers many potential advantages?

10.  What are the properties of FA that enable easier conjugation reactions?

11.  What is the process of internalization of the FA receptor–ligand complex through endocytosis?

12.  How does the targeting moiety contribute to achieving a safer drug delivery through nanocarriers?

13.  What is the synthesis of folate-conjugated porous hybrid silica nanoparticles reported by Rosenholm et al. and its purpose?

14.  What is the synthesis of FA-conjugated nanoparticles of mixed lipid monolayer shell and biodegradable polymer core for targeted delivery of docetaxel by Liu et al. and its purpose?

Regarding materials and methods:

1.     Include the manufacturer’s information for all reagents and instruments used.

Regarding figures:

1.     In Figure 1(a), include the units as a.u. or similar. 

2.     In Figure 2(b)(c), include the scale bar in the actual image in a way that can be easily visualized. 

3.     Figure 4 needs statistics (t-test, p values) to assess the significance of the statistical difference compared to the controls, otherwise no conclusions can be drawn. 

4.     All bar graphs in Figure 5 needs statistics. Panel (a) -scale bar cannot be visualized. Increase the size of the images as well. 

5.     Figure 9, panel e needs statistics. 

Author Response

RESPONSE TO REVIEWER’S COMMENTS (Reviewer #5)

The study developed folic acid-conjugated polymeric nanoparticles loaded with amodiaquine (FA-AQ NP) for cancer treatment. The nanoparticles demonstrated uniform size distribution and increased cellular uptake in various cancer cell types. The FA-AQ NP exhibited superior efficacy and antitumor ability compared to other treatments, making it a promising drug delivery system for cancer therapy.

Couple of comments for the authors:

When reading the introduction, there are a couple of questions that come to my mind that should be answered or referred within the text, including references:

Author Response: Thank you for your valuable comments. We have now modified the manuscript as per your comments.

Comments:

  1. What is the mechanism of action of amodiaquine that makes it effective against cancer?

Author Response: Thank you for your comment. Amodiaquine has been reported for its ability to inhibit autophagy, to block cell cycle progression and to induce apoptosis in different cancer types among several studies from our research group [4,6,13]. These details have now been included in the manuscript. (Lines 89-92)

  1. What are the limitations of the current chemotherapy regimen besides resistance, adverse events/toxicity, and poor targetability?

Author Response: Thank you for your comment. Current chemotherapy often requires a combination treatment strategy using support therapies to tackle side effects of chemotherapeutic agents. Overall, this strategy is expensive and makes it difficult for several patients to afford it [14]. In addition, off-target effects and poor accumulation at the tumor site are other major drawbacks. These details have been included in the manuscript. (Lines 33-36)

  1. What other nanocarrier systems, besides liposomes and polymeric nanoparticles, are developed for selective targeting against tumors?

Author Response: Thank you for your comment. Nanocarrier systems such as mesoporous silica nanoparticles, gold nanoparticles, solid lipid nanoparticles, carbon nanotubes, protein nanoparticles, core shell nanoparticles and magnetic nanoparticles have been the other systems suitable for targeting tumors [15,16]. These details have been included in the manuscript. (Lines 80-82)

  1. What is the reason for the restricted expression of folate receptors in normal tissues?

Author Response: Thank you for your comment. In normal tissue, folic acid receptors are expressed at low levels and their expression is restricted to the luminal/apical surface of polarized cells, avoiding contact with the circulation [17]. These details have been included in the manuscript. (Lines 66-68)

  1. How does the presence of high receptor affinity of FA result in high tumor tissue specificity?

Author Response: Thank you for your comment. Folic acid binds to the FR with high affinity and is capable of targeting conjugated payloads specifically to cells with overexpression of folic acid receptors. Folic acid conjugates show high affinity for the folic acid receptors, internalized into the target cell via receptor-mediated endocytosis [18]. These details have been included in the manuscript. (Lines 69-71)

  1. What are the features of FA that make it a promising targeting ligand?

Author Response: Thank you for your comment. The folic receptor is expressed at a low level in normal cells but is overexpressed during cellular activation and proliferation in human tumors. Consequently, the folic receptor is a promising target ligand for selective cancer treatment [19].

  1. How do folate receptors play a crucial role in iron homeostasis?

Author Response: Thank you for your comment. Folic acid is essential for the maintenance of cell functions and it is an essential precursor for the synthesis of purine and pyrimidine and metabolism of amino acid, thereby regulating cellular growth, proliferation, and survival [20].

  1. What are the potential advantages of using FA conjugates for targeting FRα receptors?

Author Response: Thank you for your valuable comment. FA has exhibited a specific affinity for FR receptors that are over-expressed in cancer cells compared to normal ones. Therefore, FA-conjugated prodrugs are capable of differentiating among cancerous and normal cells [21].

  1. What is the evidence that targeting FA receptors through FA offers many potential advantages?

Author Response: Thank you for your valuable comment. FA receptors are located on the luminal surface of epithelial cells in most proliferating nontumor tissues and are inaccessible to circulation. In contrast, they are expressed all over the cell membrane in malignant tissue and are readily accessible via circulation. FA receptors have the ability to bind to folic acid, a relatively innocuous, small molecule that can rapidly penetrate solid tumors and is feasible to chemical conjugation with other molecules [22]. Several studies have been reported about the efficacy of folic acid receptor targeting [18,19,21]. These details have been included in the manuscript. (Lines 63-66).

  1. What are the properties of FA that enable easier conjugation reactions?

Author Response: Thank you for your valuable comment. Folic acid is a small molecule amenable to chemical conjugation with other molecules [22].

  1. What is the process of internalization of the FA receptor–ligand complex through endocytosis?

Author Response: Thank you for your valuable comment. Internalization of the FA receptor-ligand complex through endocytosis involves four steps: (1) binding of the FA to the receptor, (2) internalization of the FA receptor-ligand complex into a membrane bound compartment, (3) dissociation of FA from the receptor and movement across the membrane into the cytoplasm, and (4) covalent addition of multiple glutamic acid residues to FA [23]. These details have been included in the manuscript. (Lines 114-118)

  1. How does the targeting moiety contribute to achieving a safer drug delivery through nanocarriers?

Author Response: Thank you for your comment. Nanocarriers improve the bioavailability and therapeutic efficiency of antitumor drugs, while providing preferential accumulation at the target site. The overall goal of utilizing nanocarriers in drug delivery is to treat a disease effectively with minimum side effects [24].

  1. What is the synthesis of folate-conjugated porous hybrid silica nanoparticles reported by Rosenholm et al. and its purpose?

Author Response: Thank you for your comment. Mesoporous silica nanoparticles functionalized by surface hyperbranching polymerization of poly(ethylene imine), PEI, were further modified by introducing both fluorescent and targeting moieties, with the aim of specifically targeting cancer cells [25]. These details have been included in the manuscript. (Lines 89-92)

  1. What is the synthesis of FA-conjugated nanoparticles of mixed lipid monolayer shell and biodegradable polymer core for targeted delivery of docetaxel by Liu et al. and its purpose?

Author Response: Thank you for your valuable comment. A system of nanoparticles of mixed lipid monolayer shell and biodegradable polymer core was developed for targeted delivery of anticancer drugs with Docetaxel as a model drug, which provide targeting versatility with a quantitative control of the targeting effect by adjusting the lipid component ratio of the mixed lipid monolayer, and combine the advantages and avoid disadvantages of polymeric nanoparticles and liposomes in drug delivery [26]. These details have been included in the manuscript. (Lines 94-99)

Regarding materials and methods:

  1. Include the manufacturer’s information for all reagents and instruments used.

Author Response: Thank you for your valuable comment. Manufacturer’s information for all reagents and instruments used have been included in the manuscript in Section 2.1 and in respective sections of method descriptions.

Regarding figures:

  1. In Figure 1(a), include the units as a.u. or similar. 

Author Response: Thank you for your comment. Unit “(a.u)” has been included in Figure 1 (a).

  1. In Figure 2(b)(c), include the scale bar in the actual image in a way that can be easily visualized. 

Author Response: Thank you for your valuable comment. Scale bar have been included in Figure 2 (b) and 2 (c) for better visualization. Updated Figure 2 has been submitted.

  1. Figure 4 needs statistics (t-test, p values) to assess the significance of the statistical difference compared to the controls, otherwise no conclusions can be drawn. 

Author Response: Thank you for your valuable comment. No significant statistical difference was observed during the stability study (Fig. 4), which indicated that there was no detrimental effect to the particle characteristics during the stability study period.

  1. All bar graphs in Figure 5 needs statistics. Panel (a) -scale bar cannot be visualized. Increase the size of the images as well. 

Author Response: Thank you for your comment. No significant statistical difference was observed between other treatments. Hence, no statistics have been included in Figure 5. Figure 5 (a) images have been enlarged for better visualization. Scale bars have been thickened for clear representation. Updated Figure 5 have been submitted.

  1. Figure 9, panel e needs statistics. 

Author Response: Thank you for your comment. No significant statistical difference was observed between other treatments. Hence, no statistics have been included in Figure 9 e.

Round 2

Reviewer 1 Report

All concerns have been addressed

Reviewer 4 Report

Authors have made responses to each questions.